# Exploring the trade-off between deep-learning and explainable models for brain-machine interfaces

**Luis H. Cubillos[1], Guy Revach[3], Matthew J. Mender[1], Joseph T. Costello[1], Hisham Temmar[1], Aren Hite[1], Diksha Zutshi[1], Dylan M. Wallace[1], Xiaoyong Ni[3], Madison M. Kelberman[1], Matthew S. Willsey[1,2], Ruud J.G. van Sloun[4], Nir Shlezinger[5], Parag Patil[1], Anne Draelos[1], Cynthia A. Chestek[1,2]**

[1]*Departments of Electrical & Computer Engineering, Biomedical Engineering, Robotics, Computational Medicine & Bioinformatics, and Neurosurgery, University of Michigan, USA*
[2]*Biointerfaces Institute and Neuroscience Institute, University of Michigan, USA*
[3]*Department of Information Technology and Electrical Engineering, ETH Zürich, Switzerland*
[4]*Department of Electrical Engineering, Eindhoven University of Technology, Netherlands*,
[5]*School of Electrical and Computer Engineering, Ben-Gurion University, Israel*

## Abstract

People with brain or spinal cord-related paralysis often need to rely on others for basic tasks, limiting their independence. A potential solution is brain-machine interfaces (BMIs), which could allow them to voluntarily control external devices (e.g., robotic arm) by decoding brain activity to movement commands. In the past decade, deep-learning decoders have achieved state-of-the-art results in most BMI applications, ranging from speech production to finger control. However, the 'black-box' nature of deep-learning decoders could lead to unexpected behaviors, resulting in major safety concerns in real-world physical control scenarios. In these applications, explainable but lower-performing decoders, such as the Kalman filter (KF), remain the norm. In this study, we designed a BMI decoder based on Kalman-Net, an extension of the KF that augments its operation with recurrent neural networks to compute the Kalman gain. This results in a varying "trust" that shifts between inputs and dynamics. We used this algorithm to predict finger movements from the brain activity of two monkeys. We compared KalmanNet results offline (pre-recorded data, $n = 13$ days) and online (real-time predictions, $n = 5$ days) with a simple KF and two recent deep-learning algorithms: tcFNN (non-ReFIT version) and LSTM. KalmanNet achieved comparable or better results than other deep learning models in offline and online modes, relying on the dynamical model for stopping while depending more on neural inputs for initiating movements. We further validated this mechanism by implementing a heteroscedastic KF that used the same strategy, and it also approached state-of-the-art performance while remaining in the explainable domain of standard KFs. However, we also see two downsides to KalmanNet. KalmanNet shares the limited generalization ability of existing deep-learning decoders, and its usage of the KF as an inductive bias limits its performance in the presence of unseen noise distributions. Despite this trade-off, our analysis successfully integrates traditional controls and modern deep-learning approaches to motivate high-performing yet still explainable BMI designs.

## 1 Introduction

Millions of people worldwide live with a neurological or spinal cord condition that limits their ability to interact with the world [1]. These patients have to rely on others for basic tasks such as eating,

moving, or even communicating with their loved ones, which severely limits their independence. Existing assistive devices, such as sip-and-puff controllers or eye trackers, can take advantage of small residual movements–if available–but are limited in the degrees of freedom (DoF) they can control. Brain-machine interfaces (BMIs) have started gaining traction in the past decades as a viable alternative that can control multiple DoFs while taking advantage of the existing hardware for computation: the brain.

Brain-machine interfaces work by reading electrical activity from the brain, decoding the activity with a decoding algorithm, and then using the algorithm's output to interact with an external device. Similar to existing applications such as language and image generation [2]–[6] and understanding [7]–[10], deep learning has become the norm in decoding algorithms for BMIs. New BMI applications such as handwriting detection from a person unable to move [11] and real-time speech production from a person unable to speak [12]–[14] have been enabled recently through the use of deep learning algorithms. Additionally, many deep learning models have demonstrated state-of-the-art results in BMI applications that previously only used linear decoders, such as real-time prediction of finger movement [15]–[17] and cursor control [18], [19].

These promising results using deep learning algorithms are enabled, in part, by an increase in the complexity of the models [20]. This can make interpreting their structure and results very difficult [10], [21], [22]. It is also difficult to predict the algorithm's behavior with out-of-distribution inputs. This complexity has caused deep learning algorithms to be seen as "black boxes," limiting their practical application in more sensitive domains such as vehicular control or lower limb prosthetics. Interaction with the physical world inevitably involves safety concerns. There, conventional control approaches that use explainable linear models are the norm, usually involving Kalman filters (KFs) because they are not as likely to exhibit unexpected behaviors. In BMIs, KFs and explainable linear models are still regularly employed in applications where the safety of the user is in question, such as when moving a robotic arm [23] or controlling a paralyzed arm through electrical stimulation [24]. Thus, researchers are confronted with the need to sacrifice potentially better decoder performance in the name of safety and explainability. Here we ask: can we have the best of both worlds?

In this paper, we examine the trade-offs between linear, explainable models and high-performing deep-learning approaches. To interpolate between these two regimes (model-based and data-driven), we modify and retrain the recently proposed KalmanNet [25], which combines the long-standing KF with the abstractness of deep models by augmenting its operation with a set of flexible recurrent neural networks (RNNs). We demonstrate the effectiveness of KalmanNet in real-time BMI experiments where monkeys with brain implants perform a dexterous finger task and show that we can match or outperform two previously published algorithms: tcFNN (non-ReFIT version) [16] and LSTM [17]. By leveraging the explainable portion of KalmanNet and its velocity dependence, we explain how this approach works through abrupt changes in the relative trust of the brain or dynamical models. Under the injection of additional noise, however, we find that while KalmanNet can reject low-magnitude noise, the LSTM actually outperforms all other models for large-magnitude out-of-distribution noise. Finally, we tested the ability of the models to generalize across task contexts and found better generalization for the simple KF than for all deep-learning-based models.

## 2 Related Work

Our work builds on previous modifications of the KF for BMI decoding, as well as advancements in deep learning and model-based techniques. Modifications to the standard KF, such as adapting it to non-linear dynamics or observation models, are common for BMI decoding. For example, Li et al. [26] used an unscented KF with a quadratic neural tuning model, achieving superior performance over the traditional KF, while others have shown that online adaptation of the KF parameters can have significant impacts on performance [27]–[29]. Additionally, modifying the trust variable of the KF–the Kalman gain–has been used to improve the computational efficiency by pre-fixing the trust value [30]. Our work further extends this concept by allowing the Kalman gain to vary over time, enabling a more flexible trust with a deep RNN. Pure recurrent and feed-forward neural networks have shown promise in BMIs. Feed-forward networks with a temporal convolution layer have outperformed the KF in an online finger control task when using the ReFIT recalibration method [16], while RNN approaches, such as [31] and [17] have achieved even better results in predicting hand and fingers position and velocities in primates. KalmanNet incorporates the benefits of these deep learning approaches with the knowledge of real-world dynamics to improve the explainability

of the model while achieving similar performance. Thus, our work also aligns with broader efforts in model-based deep learning [32], where domain knowledge is combined with deep learning to enhance performance and explainability. For example, physics-informed neural networks [33] incorporate physical knowledge into the loss function as a regularizer. KalmanNet can be framed similarly, with real-world dynamics serving as a regularizer for the underlying deep neural networks.

# 3 Methods

## 3.1 Neural Decoders

Throughout this study, we compared the performance of four neural decoders, three of which have been studied before in a BMI context: the Kalman filter (KF, [34]), the temporal convolutional feed-forward network (tcFNN [16]), and a decoder based on the long short-term memory recurrent neural network (LSTM [17]). Here, we additionally adapt a recent model, KalmanNet (KNet, [25]), to the BMI setting. Each decoder took in processed neural features and predicted movement kinematics (positions and velocities of the fingers; see Technical Appendix for more details).

**Kalman filter**. The KF algorithm tracks a state from noisy observations using a state-space (SS) description of the dynamics and the relationship between the observations and the tracked state. These are modeled as linear functions assuming Gaussian noise [35, Ch. 4]. In our study, the KF tracks a state composed of the position and velocity of both finger groups (total length of four). Like all algorithms in this study, it uses the 50ms binned spiking band power (SBP [36]) of the brain activity as observations of the state (Figure 1, B). The KF computes the level of trust in the dynamics versus the observations with a variable called the Kalman gain (Figure 1, C), and then uses it to merge both sources of information. When the linear Gaussian SS representation holds, the KF is known to achieve the optimal mean-squared error (MSE) [35, Ch. 4]. Equation 1 shows how the Kalman gain ($\mathbf{KG}$) is used to output a kinematics prediction by interpolating between the prediction given by the dynamics and the one given by the brain activity: a high value of the Kalman gain up-weights the brain activity as the main source of information, while a low value ensures the dynamics are the dominant term. The trainable parameters for the KF correspond to the linear observation model ($C$), the linear dynamics model, and the noise covariances of the state and the observations.

$$\begin{matrix} \text{Predicted} \\ \text{Kinematics} \end{matrix} = \begin{matrix} \text{Dynamics} \\ \text{Prediction} \end{matrix} \times (I - \mathbf{KG} \times C) + \begin{matrix} \text{Brain} \\ \text{Activity} \end{matrix} \times \mathbf{KG} \qquad (1)$$

**tcFNN**. The temporal convolutional feed-forward neural network (tcFNN, [16], [37]) is a "pure" (data-driven) deep learning approach that uses a temporal convolutional layer on the input, four fully-connected linear layers, and a two-dimensional output to predict finger velocities. It has been shown to predict finger velocities better than linear approaches in offline and online BMI experiments [16], [37]. The specific tcFNN architecture used throughout this paper was presented in [37] (see Technical Appendix for more details). Here, however, the ReFIT process [27], which showed the best results for online experiments in [16] was not included, to compare only the baseline performance across algorithms.

**LSTM**. The long short-term memory (LSTM [17]) network is a type of RNN that uses memory cells and gates to control what information to remember and what to forget. The LSTM has achieved state-of-the-art results in finger movement predictions, outperforming the tcFNN, the KF, and other linear and non-linear approaches [17]. The specific LSTM architecture used throughout this paper was presented in [17].

**KalmanNet**. KalmanNet [25] is a tracking algorithm that uses a (possibly mismatched) SS representation combined with deep learning [38]; It uses the KF as an inductive bias, by incorporating a deep learning architecture based on RNNs that augments the Kalman gain computation (Figure 1, C). This inclusion of the RNN allows for a more flexible modulation of trust between dynamics and observations while maintaining the linear model used to model dynamics (evolution model, $A$) and the relationship between states and observations (observation model, $C$). It differs from the KF in that it does not need an explicit model of the noise covariance of dynamics or observations, nor does it need to track the state covariance through time. The parameters of KalmanNet are the SS model parameters (necessary for the KF) and the trainable parameters of the included RNNs. Here, we modified KalmanNet to use the same dynamics and observation models that have shown success

in the KF in the past [15], [39], modified the loss function to account for the difference in scales between positions and velocities, and optimized the hyperparameters of the network for predicting finger positions and velocities.

## 3.2 Task and Data Acquisition

Two non-human primates, Monkeys N and W, were implanted with microelectrode arrays in the hand area (precentral gyrus) of the right motor cortex of the brain. Monkey N was implanted with two 64-channel arrays and Monkey W with a single 96-channel array; in both cases, recordings were limited to the best 96 channels. We recorded the SBP (power in the 300-1000Hz band) on each channel, a useful and low-power feature for BMI applications [36]. The SBP was averaged in 50ms nonoverlapping bins for each channel. For training and testing, we only used the "active" channels (i.e., those with at least one threshold crossing per second on average, approximately 20-30 total channels each day) for KalmanNet. For all other decoders, we used all channels that had threshold crossings morphologically consistent with action potentials [16], [17] (approximately 60-70 total channels, which included the "active" channels).

Both monkeys were trained to do a 2-degree-of-freedom (2-DoF) dexterous finger task in which they had to move the index and middle-ring-small (MRS) fingers independently to acquire targets shown on a screen (Figure 1, A). To successfully acquire targets, monkeys had to move their fingers to the required position and hold for a minimum hold time (750 ms). A new set of targets appeared on each trial, and the monkey had five seconds to acquire them; if the monkey could not acquire the targets during that time, the trial was labeled as unsuccessful. On *hand control* trials, finger kinematics were measured using resistive bend sensors and shown to the monkey in real-time using a custom visualization tool (MSMS [40]). On *brain control* trials, the 50-ms binned SBP was fed in real-time to a decoder that predicted finger kinematics and controlled the output visualization. The two modes—*hand control* and *brain control*–were almost identical, with the exception being the reduced hold time (500 ms) during brain control trials to lower their difficulty. Analyses of performance during hand and brain control trials are referred to as offline and online, respectively.

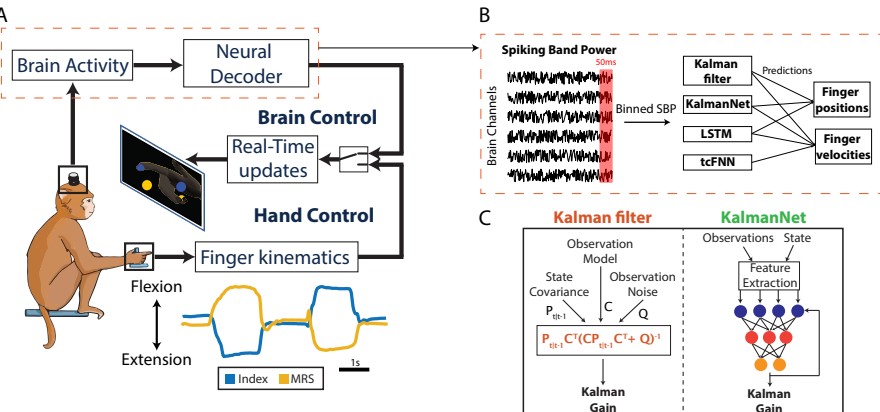

Figure 1: **Task and Neural Decoders**. (A) We trained a monkey to do a 2-DoF finger task (shown on screen) while brain activity and finger kinematics (index and middle-ring-small (MRS) traces shown below) are recorded. The monkey can do the task in hand control, using his hand, or in brain control. (B) In brain control, the SBP of each brain channel is extracted and binned every 50 milliseconds. Each neural decoder takes in a bin and predicts position and velocity, or only velocity. (C) The KF and KalmanNet differ in how they compute the Kalman gain: the KF uses the equation shown, while KalmanNet uses a set of RNNs.

## 3.3 Decoder Training and Testing

For offline decoder testing, decoders were trained in a supervised manner using a pre-recorded set of 400-500 hand-control trials from each day with synchronized brain activity and finger kinematics. The decoders were then tested on a different set of 100-500 hand-control trials from the same day to evaluate offline performance. We trained decoders daily, as micro-movements in the microelectrode

array can cause large changes in the recorded neural activity across days [41]. We trained and tested 13 different instances of each decoder overall, 8 using Monkey N's data, and 5 using Monkey W's data, and they were evaluated by their ability to predict finger positions and velocities (for KF, KalmanNet, and LSTM) or just finger velocities (for the tcFNN).

When running online (real-time trials), decoders were trained on a similar initial set of 400-500 hand-control trials and then tested in pairs each day in an ABA manner: decoder 1 for a set of trials, decoder 2 for another set, and then back to decoder 1 for another set, to account for possible shifts in signals or monkey behavior. All decoders were tested using a similar number of trials in each set. KalmanNet was compared online with the KF on four days, with the tcFNN on two days (during which the KF was also run), and with the LSTM on one day. All days on which the monkey attempted to do online trials with KalmanNet and at least one more decoder were included in our analysis. Only Monkey N was used for online experiments.

### 3.4 Performance Metrics

We used different metrics to determine decoder performance when tested offline versus online. When testing a decoder offline, we have access to the ground-truth finger kinematics and thus can directly measure prediction accuracy. We used the Pearson correlation coefficient and the MSE between the ground truth and the predictions and then averaged across fingers to get a single value for each position and velocity. To compare offline performance across models, we used a combination of frequentist and Bayesian statistical methods. First, paired-sample t-tests on the MSEs as well as in the Fisher Z-transformed correlation coefficients [42] determined whether the difference was significant ($\alpha = 0.05$) under the null hypothesis assumption. Second, we computed the Bayes factor ($B_{01}$) to determine the ratios of likelihoods between the null and alternative hypotheses.

When testing a decoder online, without access to ground-truth finger kinematics, we can simply measure how well the monkey completed the trials. We used four metrics for online performance: success rate, trial times, path efficiency, and throughput. The success rate is the ratio of the number of trials successfully completed to the number of trials attempted. Trial time is the time taken to acquire both finger targets minus the hold time: higher trial times mean a slower or more erratic movement. The path efficiency measures the smoothness of the movements by computing the ratio between the distance between the start position and the target and the distance traveled by the monkey when completing the trial: higher path efficiencies mean smoother paths. Finally, the information throughput (measured in bits/s) accounts for each trial's difficulty (distance between start and target) and completion time [16], with a higher throughput representing faster target acquisitions. Note that all metrics except the success rate were computed only with successful trials. Since there is some variability in performance across days due to micro-movements of the electrode array [41], we normalized all the metrics–except success rate–from each online day by the average performance of KalmanNet during that day, to be able to aggregate each decoder's performance across testing days.

### 3.5 KalmanNet as a Non-linear Trust System

To understand the mechanism used by KalmanNet to accurately predict finger kinematics, we inspected the network's output, which corresponds to the Kalman gain matrix in the KF. For easier interpretation, we computed the Frobenius norm of KalmanNet's Kalman gain and compared it to the norm of the predicted velocities. As a metric of the relationship, we used the Pearson correlation coefficient between the Frobenius norm of the Kalman gain and the vector norm of the velocities over time across all 13 offline days tested for both monkeys ($n = 8$ for Monkey N, $n = 5$ for Monkey W).

Additionally, to replicate KalmanNet's observed correlation between Kalman gain and velocity in the regular KF, we modified the KF to have heteroscedastic process noise. We refer to this model as the heteroscedastic Kalman filter (HKF). The process noise determines how noisy we believe the dynamics to be: with noisier dynamics, the KF will trust observations more (higher Kalman gain), and with less noisy dynamics, the KF will trust the dynamics more (lower Kalman gain). Thus, to force the Kalman gain to covary with the velocity, we made the process noise, specifically the parts related to the velocities, increase linearly with the norm of the ground truth velocity (see Technical Appendix for more details). Note that this model needs the ground truth velocity to properly vary the process noise, which makes it unsuitable for online experiments.

### 3.6 Robustness to Real-World Operation

To test the models' offline robustness to real-world operation, we first simulated noise spikes on the data. This may happen in a real-world scenario with, for example, movement artifacts or loose electrical connections, and we investigated whether this would have unsafe consequences when controlling a physical device, such as big spikes in the velocity predictions. The noise spikes were simulated by adding a fixed magnitude to all neural channels of the testing dataset for a given number of 50-millisecond bins. We varied the magnitude from 0.1 to 100 times the standard deviation of the training set SBP and varied the duration from one to five 50-millisecond bins. While this may seem like a large amount of noise, for BMI applications, it is possible to have unexpected $mV$ level noise added to $\mu V$ level neural signals. The noise was injected randomly for 5% of the total time steps on each testing day, and we tested it offline across 13 different days for both monkeys. We measured the impact of the noise in the models with the MSE of the velocity normalized by the baseline velocity MSE (no noise added to the testing dataset).

Additionally, we tested the decoders' ability to generalize to unseen task contexts, a key property for decoders to work in real-life scenarios rather than just in highly constrained research settings [37], [43]. On six different days, we modified the task by having the wrist flexed by 30 degrees and adding a spring to resist flexion and facilitate extension [43]. Then, we trained off-context and on-context decoders. Off-context decoders were trained on a normal set of trials (no wrist or spring modifications) and then tested on the last 20% of the spring+wrist context trials. On-context decoders were trained on the first 80% of the spring+wrist context trials and then tested on the last 20% (see Figure 6, A). We evaluated the generalization performance of each algorithm by measuring the percentage increase in velocity MSE from the on-context to the off-context decoders.

## 4 Results

### 4.1 Offline and Online Performance

To determine how well KalmanNet could fit the data compared to state-of-the-art deep learning techniques, we first assessed its offline performance in predicting finger kinematics from brain data. We evaluated KalmanNet over $n = 13$ days across the two monkeys, comparing it with the KF and the deep learning based tcFNN and LSTM models.

The LSTM was the highest-performing approach in a previous study [17]. In terms of correlation with velocity, which controls the visualization when running online, KalmanNet did not significantly differ from the LSTM ($p = 0.64$, $B_{01} = 3.3$) and had a significantly higher correlation than the KF ($p < 1e-7$, $B_{01} = 0.048$) and the tcFNN ($p < 0.001$, $B_{01} = 0.062$; Figure 2, B). All approaches had similar correlations with position, other than the tcFNN, which is a velocity-only approach. Regarding velocity MSE, there was also no significant difference between KalmanNet and the LSTM ($p = 0.72$, $B_{01} = 3.4$) or between KalmanNet and the tcFNN ($p = 0.29$, $B_{01} = 2.1$). However, KalmanNet significantly outperformed the KF ($p < 0.01$, $B_{01} = 0.14$). In terms of position, there were no significant differences in correlation (KNet vs. LSTM, $p = 0.32$, $B_{01} = 2.4$; KNet vs KF, $p = 0.93$, $B_{01} = 3.5$; LSTM vs. KF, $p = 0.08$, $B_{01} = 1.1$) or MSE (KNet vs. LSTM, $p = 0.26$, $B_{01} = 2.0$; KNet vs KF, $p = 0.77$, $B_{01} = 3.5$; LSTM vs. KF, $p = 0.1$, $B_{01} = 1.1$), although KalmanNet had higher variance in MSE. Overall, it is important to note that the absence of p-values below the significance level for position and velocity correlations and MSE between the LSTM and KalmanNet should not be interpreted as definitive evidence that the two models are equivalent. Instead, it indicates that the data did not provide strong enough evidence to conclude that the models were different. However, the Bayes factor ($B_{01}$) values between 2 and 3.4 suggest that the models may have comparable performance, with the null hypothesis (models are equivalent) being at least twice as likely as the alternative hypothesis (models are different). This suggests a higher likelihood that any observed differences are not substantial, though it remains essential to consider the variability and sample size when interpreting these results.

However, because this is a motor control application, it is important to evaluate performance online, with a user in the loop. Therefore, we tested all algorithms online across five sessions (number of trials: KF = 601, tcFNN = 576, KNet = 2801, LSTM = 393) and compared their performance in terms of success rate, throughput, trial times, and path efficiencies. Figure 3 (right) shows the performance of all approaches normalized to the performance of KalmanNet (except for the success

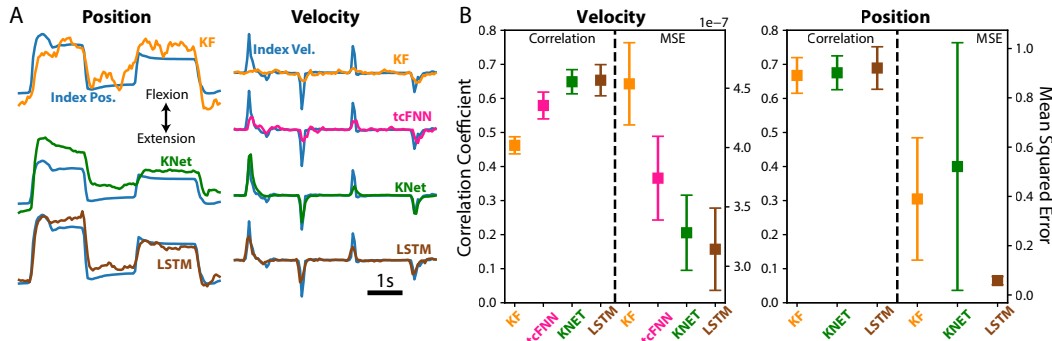

Figure 2: **Offline Performance**. (A) Traces of ground truth index position and velocity (in blue) versus the predictions from each neural decoder. Note that tcFNN only predicts finger velocity. (B) Velocity (left) and position (right) performance in terms of correlation and MSE for each neural decoder. Square markers and error bars denote the mean and the standard error of the mean, respectively. Tested across $n = 13$ days from both monkeys.

rate). We found that KalmanNet outperformed the KF and tcFNN across all metrics (Figure 3, Supplemental Videos 1 and 2, respectively), allowing the monkey to perform faster and more accurate trials. Compared to the LSTM, KalmanNet achieved lower throughputs ($p < 1E - 16$) and higher trial times ($p < 1E - 5$), meaning slower trials overall. However, it had higher overall success rates (97% vs. 90%) and more efficient and smoother paths ($p < 1E - 7$; Figure 3; Supplemental Video 2), which is consistent with the inclusion of KalmanNet's dynamics model. The online example traces tell a similar story, with the KF and tcFNN showing more oscillations in the path and more unsuccessful trials (Figure 3, A).

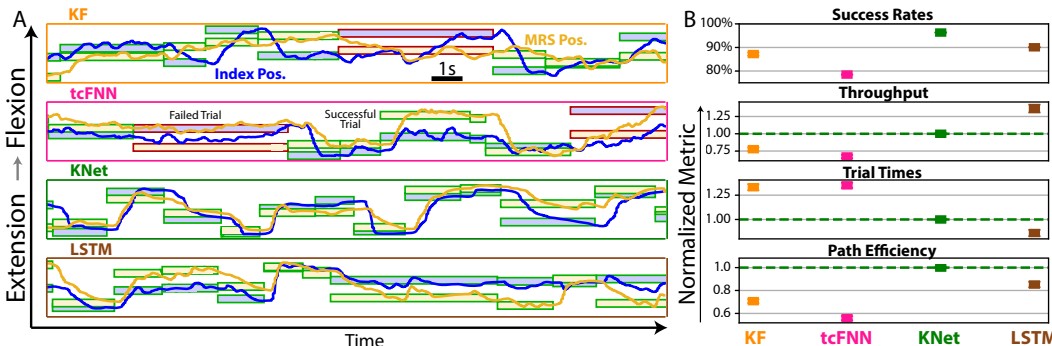

Figure 3: **Online Performance**. (A) Traces of the positions of the index (blue) and middle-ring-small (yellow) fingers during online control, across neural decoders. Blue-filled boxes represent index targets in the flexion-extension range, while yellow-filled boxes represent MRS targets. Green-outlined boxes represent successful trials, while red-outlined boxes show failed trials. (B) Online metrics of performance. Throughput, trial times, and path efficiencies are normalized to the corresponding KalmanNet value for each day. Tested on monkey N across $T = 601$ (KF), 576 (tcFNN), 2801 (KNet), 393 (LSTM) trials in a total of five days.

## 4.2 Modulating Trust between Dynamics and Observations

Notably, KalmanNet can match or nearly match the highly nonlinear black-box LSTM performance. In KalmanNet, the structure of the linear KF is intact, such that the performance improvement comes from the RNNs used to compute the Kalman gain. This is the KF's metric of trust that determines how to interpolate between dynamics and observations (see Methods). To analyze this, we first inspected the regular KF's Kalman gain and observed that it converges to a fixed value in the first few seconds of a session (inset in Figure 4, A, orange line), in line with its known steady-state behavior [35, Ch. 4.3]. We then evaluated the KalmanNet gain computation during inference. Interestingly, we found

that it correlated strongly with KalmanNet's output velocity, both offline (mean + standard error of the mean = $0.79 \pm 0.02$) and online ($0.60 \pm 0.02$; Figure 4, center). This correlation suggests that KalmanNet behaved as a non-linear trust system, switching rapidly towards trusting the neural activity when the velocity was high and quickly switching to trusting the dynamics when it needed to stop.

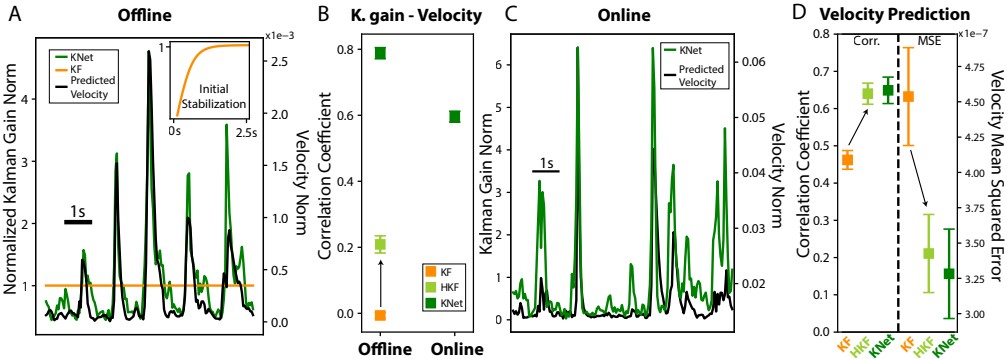

Figure 4: **KalmanNet as a Non-linear Trust System**. (A) Offline trace showing the norm of the Kalman gain for KalmanNet (in green) and for the KF (in orange), together with the predicted velocity for KalmanNet (black). The inset shows the first five seconds of trials, during which KF's Kalman gain converges to a fixed value [35, Ch. 4]. (B) Average correlations between Kalman gain and velocity for KalmanNet, HKF, and KF, for offline ($n = 13$ days, across both monkeys) and online ($n = 5$ days, only Monkey N) trials. (C) Online trace of the norm of the Kalman gain for KalmanNet (green), together with the predicted velocity (black). (D) Offline velocity prediction performance comparison between KF, HKF, and KalmanNet, in terms of correlation coefficient (left) and MSE (right). Arrows represent the addition of a heteroscedastic process noise to transform the KF into HKF. Tested across $n = 13$ days and both monkeys.

To further explore whether this observed modulation of trust is the key to KalmanNet's improved performance, we used our knowledge of the true velocity to manipulate the noise covariance of the regular KF to achieve the same effect. Specifically, we created a new KF model with heteroscedastic process noise that increased with high velocities and decreased with low velocities, forcing the Kalman gain to covary with the output velocity (HKF model; Kalman gain correlation with velocity output across days and monkeys = $0.21 \pm 0.03$; Figure 4, B). We found that this modification made the new model perform as well as KalmanNet when predicting velocity, both in terms of correlation (HKF = 0.64 vs. KNet = 0.65, $p = 0.57$) and MSE (HKF = $3.4E - 7$ vs. KNet = $3.3E - 7$, $p = 0.73$), which showcases the benefits of the trust modulation mechanism observed in KalmanNet.

### 4.3 Robustness to Noise and Change in Task Context

The primary motivation for using a principled KF rather than a black box neural network is its safe operation, even under new or changing circumstances. The KF conventionally leverages physics-based SS models to improve kinematic predictions from noisy sensors. Therefore, we evaluated these approaches in the face of artificially-injected sensor noise of varying magnitudes and duration across channels (see Methods). This allowed us to study the effect on the output velocity under various real-world scenarios. Given the small magnitude of brain signals, noise artifacts can be much larger than the signal features of interest. We modeled those with extreme noise additions 100 times the standard deviation in the training data. Surprisingly, KalmanNet had the largest increase in MSE associated with injected noise (by a factor of $\sim 4300$ for noise on three consecutive time bins, when the noise magnitude was 100x the std. dev.). It exhibited high-velocity spikes at the noise injection points (Figure 5 A and Supplemental Figure 9). Perhaps surprisingly, the best performer was the LSTM, which changed very little from the no-noise case. It showed just a $\sim 2\times$ increase in velocity MSE when injected with a noise level 100 times the training data's standard deviation. It is important to note that the noise injected was not present in the training data, so even the regular KF performed poorly. However, unmodeled noise is a common scenario in BMIs stemming from, for example, problems with the physical connection to the recording electrodes or electrode movement with respect to neurons [41].

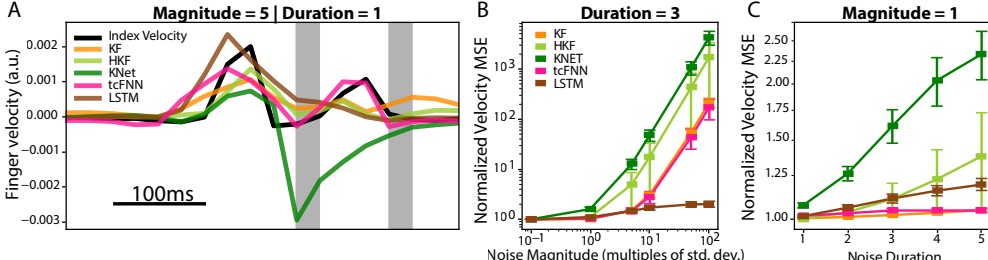

Figure 5: **Robustness to Injected Noise**. (A) Offline traces across neural decoders as compared to the ground truth index velocity (black). Grey columns represent noise injections of 5 times the standard deviation of the training dataset distribution, for a duration of one time bin. (B) Change in normalized velocity MSE as the noise magnitude changes, for a fixed noise duration of 3 time bins. (C) Change in normalized velocity MSE as the noise duration changes, for a fixed noise magnitude of $1\times$ the training dataset distribution's standard deviation. Tested on 13 days for Monkeys N and W.

Another potential benefit of a KF with a simple linear model is the ability to generalize to new contexts. Therefore, we evaluated all decoders by slightly varying the monkey task. We trained all models on the normal task context and then tested them in a new context [37], [43] (off-context performance), which contained a spring to resist flexion, as well as a 30-degree angle change in wrist position (Figure 6, A). We then compared their performance to training and testing in the new context (on-context performance). The normal KF showed the lowest drop in absolute performance (7.98% MSE), exhibiting good generalization to the new model but still showing the lowest absolute performance. On the other hand, all neural network models, including KalmanNet, showed substantially higher MSE increases due to the context shift (34.20% for tcFNN, 44.45% for KNet, 50.34% for LSTM).

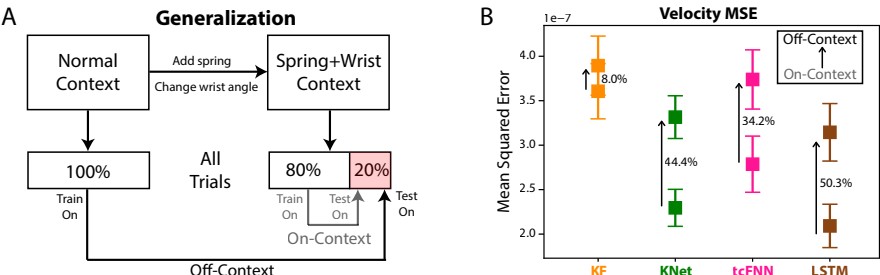

Figure 6: **Generalization Across Task Contexts**. (A) Illustration of how off-context and on-context decoders are trained. Monkeys performed a variation of the task that included a change in wrist angle and a spring that resisted flexion. Off-context decoders were trained on a normal set of trials from the same day and then tested on the last 20% of the trials of the spring+wrist context. On-context decoders were trained and tested on the same spring+wrist context. (B) Change in velocity MSE between on and off-context decoders for each decoder. Tested across $n = 6$ days of Monkey N.

## 5 Discussion

In this paper, we explored the trade-offs between explainable linear models and deep-learning approaches to BMI decoding. We modified and retrained KalmanNet [25], which fuses deep learning techniques with the classic KF. We showed that KalmanNet can match or outperform existing state-of-the-art decoders in offline and online modalities. We also showed that the explainability of KalmanNet can be leveraged to improve a linear model up to the state-of-the-art standard but that it comes at the cost of greater error when faced with out-of-distribution noise.

KalmanNet approached the state-of-the-art performance offline, achieving comparable velocity and position predictions to the LSTM [17]. This performance does not necessarily transfer to online trials, as has been studied previously [19], [44], and as we observed here with the tcFNN model. Offline, the tcFNN had significantly better velocity predictions than the KF, but online, it had overall worse

success rates, throughput, trial times, and path efficiencies. This lower online performance may follow from the fact that the monkey is in the loop, reacting in real-time to the output of the decoder, which may change the dynamics of the problem versus testing decoders offline [19], [44]. Note, however, that we did not implement ReFIT [16], [27] for tcFNN, a recalibration technique that can improve tcFNN online performance [16]; a similar method could potentially be developed for all decoders (albeit with potential differences in performance improvements), and thus we decided to compare all models with their baseline non-ReFIT performances (see Technical Appendix). KalmanNet, on the other hand, was able to work well online and did similarly well to the LSTM: it acquired targets more slowly but had higher success rates as well as smoother paths, which may be more desirable for applications controlling physical devices. Moreover, while we currently use existing SS modeling that is learned separately from the filtering task, the fact that KalmanNet converts the KF into a machine-learning model indicates that one can potentially learn the observation model ($C$) and its computation of the Kalman gain. As demonstrated in [45], this could lead to both improved MSE performance and provide information on the usefulness of existing approaches for forming SS representations in BMI.

KalmanNet functioned as a non-linear trust system, quickly re-weighting the contributions to its predictions from either the model inputs or internal dynamics depending on the desired output. For stopping, since the dynamical model provides an explicit term for exponentially decreasing the output velocity (see Technical Appendix), KalmanNet learned to trust the dynamical model. For fast movements, only the neural activity contains this information. Therefore, it chose to put more confidence in the neural inputs. We constructed a new linear model that enforced this observed behavior (HKF), and it matched offline state-of-the-art results, albeit assuming that the velocity level is known. This showcases some of the potential benefits of using a more explainable and, therefore, trustworthy model compared to a black-box decoder. Additionally, the smart switching between dynamics and inputs is an interesting behavior that could be enforced even in black-box deep-learning decoders to potentially improve decoding. Similar approaches were recently explored for speech neuroprostheses, using a combination of a language model (which can be thought of as language "dynamics") and an RNN predicting phonemes from neural activity [14]. One could modify that approach by, for example, having a model that learns to trust the neural activity for starting words or when faced with uncommon transitions between phonemes, and to trust the language model on more common phoneme transitions or for being silent. This approach would be similar to the behavior we observed in KalmanNet.

A disadvantage of combining a neural network with a Kalman filter is that the overall method carries over some limitations from both approaches. We found that when faced with out-of-distribution noise, KalmanNet had the highest increases in MSE, up to $1000\times$, when faced with noise $100\times$ the original distribution's noise. This effect likely follows from the similar behavior exhibited by the KF, which increased its MSE $100\times$ with the same noise magnitude. Nonetheless, the fact that KalmanNet preserves the operation of the KF indicates that it can be extended to outlier-robust variations of the KF, e.g., [46], or alternatively, combined with hypernetworks for handling multiple noise levels [47]. Interestingly, the pure RNN approach (the LSTM) had the lowest increase in error at high noise magnitudes. This result may be unexpected for control-focused researchers, who often choose linear approaches due to the perceived higher safety of the more explainable models.

In conclusion, the results shown here with KalmanNet suggest that we may not need to sacrifice explainability to achieve results in line with the state-of-the-art deep-learning models and that using a mechanism based on gating trust between dynamics and observations may provide valuable insight for developing future models.

## 6   Acknowledgements

We thank Eric Kennedy for animal and experimental support and Professor Hans-Andrea Loeliger for helpful discussions. This work was supported by NSF GRFP 1841052, NSF EFRI BRAID 2223822, CASI award (1021865.01) from the Burroughs Wellcome Fund, NSF NCS 1926576, NIH R01-NS-105132-01-A1, NIH U41 NS129436 and the Agencia Nacional de Investigacion y Desarrollo (ANID) of Chile.

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

# A  Technical Appendix

## A.1  Kalman filter

### A.1.1  Structure

The regular Kalman filter was first introduced by Kalman [34] and provides a reliable way of tracking state variables over time, given periodic sensor measurements that contain some information about the state. In our case, the state to track is composed of the kinematics of both finger groups:

$$x = [p_{\text{idx}} \ p_{\text{mrs}} \ v_{\text{idx}} \ v_{\text{mrs}}] \tag{2}$$

Where $p_i$ represents the flexion angle in arbitrary units (from 0 to 1) for finger group $i$, while $v_i$ represents the flexion velocity in arbitrary units/s for finger group $i$. The sensor measurements, in our case, correspond to the spiking band power (SBP) of the neural channels. Thus, an observation at time $k$ would be:

$$y_t = [y_1 \ y_2 \ ... \ y_n] \tag{3}$$

Where $n$ corresponds to the number of neural channels used during the experiment. The KF predicts a new state by first computing the Kalman gain, which uses information about the noise covariance of the dynamics ($W$) and observations ($Q$) to determine the optimal interpolation between dynamics and observation predictions (equation 6).

$$x_{t|t-1} = Ax_{t-1|t-1} \tag{4}$$
$$P_{t|t-1} = AP_{t-1|t-1}A^T + W \tag{5}$$
$$K_t = P_{t|t-1}C^T(CP_{t|t-1}C^T + Q)^{-1} \tag{6}$$
$$x_{t|t} = (I - K_tC)x_{t|t-1} + K_ty_t \tag{7}$$

In equations 4 through 7, $A$ represents the linear dynamical model that encodes the natural evolution of the state through time, $C$ the linear observation model that relates the observations to the state, and $P_{k|k}$ the covariance of the state prediction. Equation 7 uses those models, together with the Kalman gain, to compute a prediction of the state at time $t$, $x_{t|t}$. When running the Kalman filter online, since the monkey can see in real-time the position of each finger group of the virtual hand, we set the covariance of the position estimate to zero, which has shown to improve results previously [15], [39]. Additionally, to control the virtual hand on the screen in online trials, we calculated its position using the following equation: $\text{pos}_t = 0.02 \cdot \hat{\text{pos}}_t + 0.98(\text{pos}_{t-1} + 50 \cdot \hat{\text{vel}}_t)$, which allowed us to interpolate between the position predicted directly ($\hat{\text{pos}}_t$), and the position predicted through the velocity ($\hat{\text{vel}}_t$). The velocity term allows for smooth movements, while the position term helps to stabilize the overall position prediction, and the specific values were found empirically.

### A.1.2  Model Training

The Kalman filter parameters (dynamics model $A$, observation model $C$, and noise covariances $Q$ and $W$) were computed using a combination of domain knowledge and a data-driven approach. The linear dynamics model $A$, which represents the evolution of the state through time, had the structure shown in equation 8 [39]. The model integrates velocity (with dt = 50ms being the bin size) and assumes no relationship between positions at time $t-1$ and those at time $t$, as well as no relationship between velocities at time $t-1$ and velocities at time $t$. The velocity transition sub-matrix, represented in equation 8 with $A_{\text{vel}}$, was learned from an initial calibration run (see Methods) using a least-squares approximation. Note that across days, the diagonal elements of $A_{vel}$ were always strictly greater than 0 and lower than 1, representing an exponential attenuation of the velocity through time: if no input is applied to the movement, the dynamical model allows for fast stopping.

$$A = \begin{bmatrix} 1 & 0 & dt & 0 \\ 0 & 1 & 0 & dt \\ 0 & 0 & & \\ 0 & 0 & A_{\text{vel}} & \end{bmatrix} \tag{8}$$

The observation model $C$ was computed in a similar way. The only difference is that no assumptions were made about the structure, and a bias term was added to account for the difference in base levels. Then, given $A$ and $C$, we computed the noise covariance for the dynamics ($W$) and the observations ($Q$) using the maximum likelihood estimation. For $W$, we additionally set the noise covariance of the position terms to zero, assuming that the error is only propagated through the velocity term [39].

In the training of Kalman filter model, an ASUS TUF A15 laptop equipped with an NVIDIA 1660 Ti GPU was utilized. The model took approximately 10 seconds to train for each day.

## A.2   KalmanNet

### A.2.1   Structure

The main difference between KalmanNet and the Kalman Filter is that KalmanNet uses a set of recurrent neural networks (RNNs) to compute the Kalman Gain. In our implementation, KalmanNet uses the same linear dynamics ($A$) and observation ($C$) models, but now does not require an explicit definition of the noise covariances. The full architecture, explained in more detail in [25], is shown in Figure 7. Briefly, it attempts to compute the Kalman gain by using gated recurrent units (GRUs, a type of recurrent neural network) to implicitly model the noise covariance of the dynamics (GRU 2 in Figure 7), the noise covariance of the observations (GRU 1 in Figure 7), and the prediction covariance of the observations (GRU 3 in Figure 7. This implicit modeling of the noise covariances allows KalmanNet to dynamically shift the Kalman gain depending on the current state and inputs to choose a better interpolation between states and observations. Overall, the set of equations for KalmanNet is reduced to equations 4 and 7, with $K_t$ corresponding to the output of the network at time $t$.

The inputs to the KalmanNet network are variations on the Kalman filter variables: the observation difference ($F_1$, eq. 9), the innovation difference ($F_2$, eq. 10), the forward evolution difference ($F_3$, eq. 11), and the forward update difference ($F_4$, eq. 12) [25].

$$F_1 = y_t - y_{t-1} \tag{9}$$
$$F_2 = y_t - C x_{t|t-1} \tag{10}$$
$$F_3 = x_{t-1|t-1} - x_{t-2|t-2} \tag{11}$$
$$F_4 = x_{t-1|t-1} - x_{t-1|t-2} \tag{12}$$

When running online, the position of the virtual hand was calculated as for the Kalman filter, integrating the velocity prediction and using 2% of the position prediction for stabilization.

### A.2.2   Model Training

To train the KalmanNet network, the loss function used was a scaled mean-squared error (MSE) that accounted for the differences in scale between position and velocity, computed between the predicted and ground truth output. The network does not produce the predicted state directly; instead, it generates the Kalman gain, for which we do not have a ground truth. Consequently, for each time step, a standard update and prediction procedure was executed, incorporating the KalmanNet network output as a component of the computation (using equations 4 and 7). Sequences of sixty time bins (equivalent to three seconds of data) were passed through the network, and then the loss was computed as the average scaled MSE between the output at each time step and the ground truth state from the training data.

The KalmanNet network for each day was trained for 300 iterations, each taking a batch of 16 random sequences of length 60 in the training data (Figure 8). Since to produce an output, KalmanNet needs to do a forward pass on the network, as well as a Kalman filter step, most of the pyTorch optimizations for network training could not be used, which slowed down training. Overall, the network training

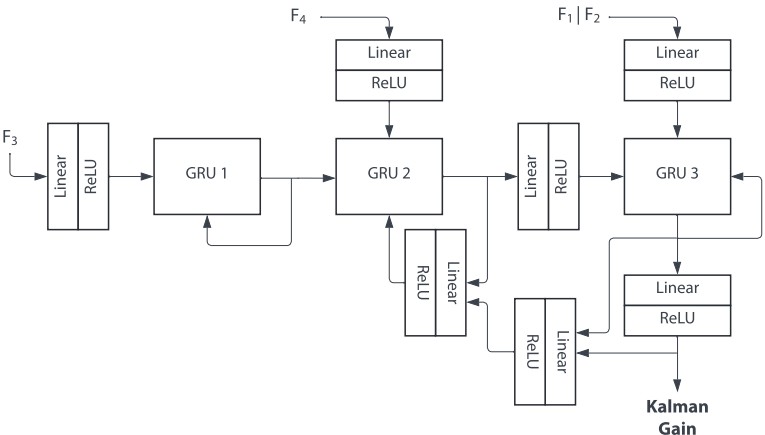

Figure 7: **KalmanNet architecture** Diagram of the components of the KalmanNet network. It consists of three GRUs plus seven linear + ReLU layers that try to model the normal way of computing the Kalman gain [25]. $F_1$ through $F_4$ correspond to the input features from equations 9 through 12.

took around 10-15 minutes per day on an Asus TUF A15 laptop equipped with an NVIDIA 1660 Ti GPU.

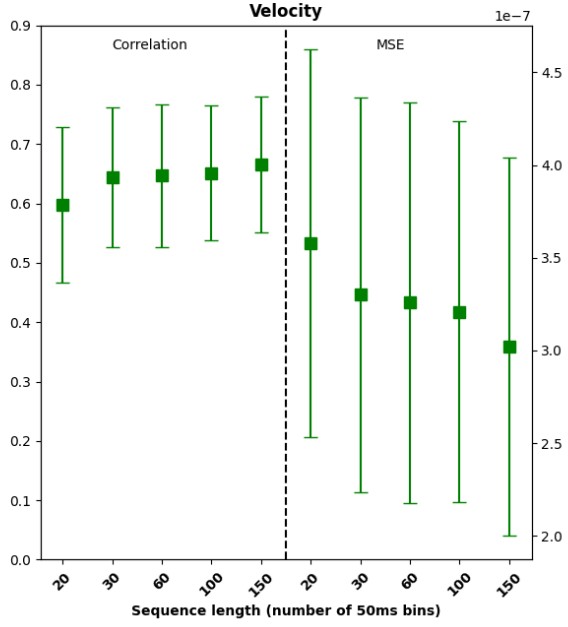

Figure 8: **Sensitivity analysis of sequence length during training**. Offline velocity correlation (left) and MSE (right) for KalmanNet, under different sequence lengths employed during training. The horizontal axis represents the number of 50ms bins; the one used throughout corresponds to 60, or equivalently, three seconds. Computed across all $n = 13$ days for both monkeys.

## A.3 Heteroscedastic Kalman filter

The heteroscedastic Kalman filter (HKF) is equivalent to the KF in all but the process noise covariance ($W$). To enforce a behavior similar to what we observed with KalmanNet's output covarying with the

velocity, we forced the dynamics noise covariance in the HKF to also covary with velocity. At time $t$, and given a base dynamics noise model $W$ (shown in equation 13), we modified the velocity-related terms using the ground truth velocity of index and MRS at time $t$, $v_t$, to produce a new time-synced noise model $W_t$ (equation 14). In the new noise model, $\alpha$ corresponds to a constant scaler and $v_{\max}$ to the maximum absolute velocity observed across all experiments.

$$W = \begin{bmatrix} 0 & 0 & 0 & 0 \\ 0 & 0 & 0 & 0 \\ 0 & 0 & \sigma^2_{v^{\text{idx}}} & \sigma_{v^{\text{idx}}}\sigma_{v^{\text{mrs}}} \\ 0 & 0 & \sigma_{v^{\text{idx}}}\sigma_{v^{\text{idx}}} & \sigma^2_{v^{\text{mrs}}} \end{bmatrix} \tag{13}$$

$$W_t = \alpha \begin{bmatrix} 0 & 0 & 0 & 0 \\ 0 & 0 & 0 & 0 \\ 0 & 0 & \sigma^2_{v^{\text{idx}}}\frac{|v_t^{\text{idx}}|}{v_{\max}} & \sigma_{v^{\text{idx}}}\sigma_{v^{\text{mrs}}}\frac{||v_t||}{v_{\max}} \\ 0 & 0 & \sigma_{v^{\text{idx}}}\sigma_{v^{\text{mrs}}}\frac{||v_t||}{v_{\max}} & \sigma^2_{v^{\text{mrs}}}\frac{|v_t^{\text{mrs}}|}{v_{\max}} \end{bmatrix} \tag{14}$$

Note that since we are changing the noise model, rather than the Kalman gain directly, the output Kalman gain of the HKF does not perfectly track the velocity, as the Kalman gain takes a few steps to account for changes in the noise model.

## A.4  tcFNN

The tcFNN stands for temporally convolved Feedforward Neural Network, and was introduced by [16] and then further studied by [37]. It consists of an initial time convolutional layer that takes in three time bins (150ms) of neural activity followed by four linear layers using ReLU activation functions, batch normalization, and 50% dropout. Note that tcFNN only predicts velocity outputs, rather than position and velocity, as the other decoders tested in this study do. See [16] and [37] for more details on architecture and implementation.

The tcFNN was trained for 15 epochs using the hyperparameters used in [37], and the training took around 1 minute per day on an Asus TUF A15 laptop equipped with an NVIDIA 1660 Ti GPU.

When running online, the position of the virtual hand was calculated by integrating the velocity prediction since no position prediction was available. Additionally, note that previous work with tcFNN has shown that implementing a recalibration feedback intention-training (ReFIT) step after the initial training can substantially improve online performance by assuming that errors during online trials are mostly due to errors in decoder output. In this work, we decided to compare to the base (non-ReFIT) tcFNN, as ReFIT has also been shown to greatly improve performance in the Kalman filter and a similar concept could also be implemented for recurrent approaches, such as the LSTM and the KalmanNet network.

## A.5  LSTM

The long short-term memory (LSTM) network is a type of recurrent neural network that has more control over what to remember and what to forget about long-term relationships between inputs. In this study, we refer to LSTM as the architecture used in [17], consisting of a single-layer LSTM with a linear layer mapping its hidden output to position and velocity predictions. See [17] for more details on architecture and hyperparameters.

The LSTM used the same hyperparameters as in [17] and was trained with a scheduler that adapted the learning rate according to the validation loss, and stopped after the validation loss stopped improving. The training took around 2 minutes per day on an Asus TUF A15 laptop equipped with an NVIDIA 1660 Ti GPU.

When running online, the position of the virtual hand was calculated as for KalmanNet and the Kalman filter, integrating the velocity prediction and using 2% of the position prediction for stabilization.

# B  Supplemental Figures

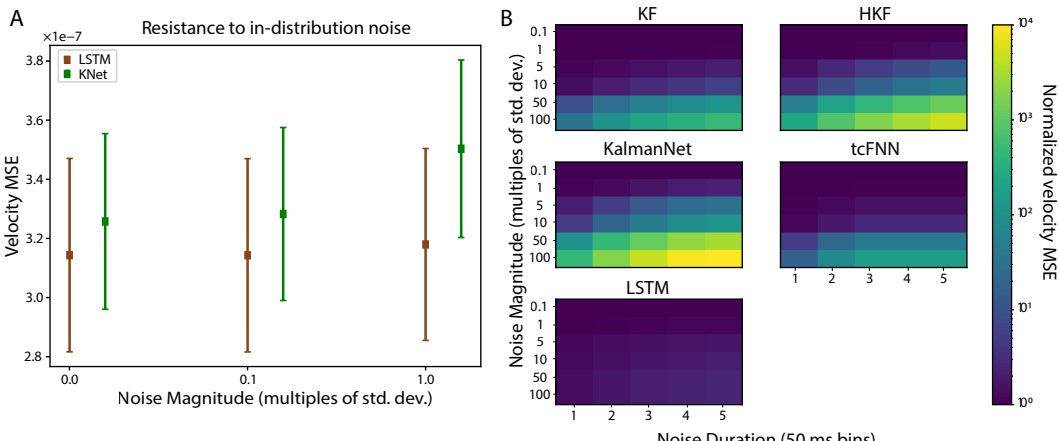

Figure 9: **Resistance to noise injection**. (A) Offline velocity MSE for KalmanNet (green) and LSTM (brown) across n=13 days for both monkeys, with noise values closer to those present in the training data. A noise of zero magnitude is equivalent to not adding noise (i.e., baseline shown in Figure 2). (B) Full product of normalized velocity MSE across models for all values of noise magnitude and duration. The logarithmic color bar on the right represents the MSE value for each combination of noise magnitude and duration, normalized to each model's baseline performance (without noise).

