# OpenReview forum: "Exploring the trade-off between deep-learning and explainable models for brain-machine interfaces"
_NeurIPS.cc/2024/Conference — NeurIPS 2024 poster_

### Official Review · Reviewer_xWsH · 2024-06-20

**Soundness:** 3
**Presentation:** 3
**Contribution:** 3
**Rating:** 6
**Confidence:** 3

**Summary:**

The study provides a rigorous comparison of four neural decoders typically used in BCIs: Kalman Filter, KalmanNet, tcFNN, and LSTM in both offline and online conditions on a NHP performing a 2 degree of freedom dexterous finger task. Authors explore the trade-off in decoding capabilities of these decoders with their explainability. Authors find that the KalmanNet which incorporates deep-learning techniques by estimating the Kalman Gain through GRUs can attain comparable performance to fully "Black-Box" methods using LSTMs, showing there exists methodology which can still retain the performance of deep-learning techniques without fully sacrificing the explainability of linear models. Furthermore, authors also analyze the behavior of the GRUs of the KalmanNet by analyzing the corresponding Kalman Gain, and show that the behavior of KalmanNet can be replicated by a explainable heteroskedastic Kalman Filter. Authors also discuss some limitations of these deep learning approaches (including KalmanNet) when dealing with out-of-distribution inputs.

**Strengths:**

- The paper is well-written and easy to follow.
- The paper tackles an important problem of understanding the trade-off between the performance of "black-box" decoders with superior decoding abilities and simpler linear decoders  with more explainability in the context of BCI.
- The authors provide rigorous comparison between 4 different decoders with varying levels of "black-box-ness" starting from the conventional linear Kalman Filter, to KalmanNet (which introduces non-linearity by estimating the Kalman Gain using a GRU architecture), and two conventional deep-learning approaches: LSTM & tcFNN on both online and offline data collected from NHPs.
- The authors are able to show that it is possible to adapt traditional linear approaches by incorporating deep learning approaches (i.e., KalmanNet) while still retaining some amount of explainability without sacrificing the superior performance.
- Authors are also able to explain the KalmanNet using the explainable Kalman Filter with heteroscedastic noise models.
- Authors also demonstrate that deep learning approaches (including KalmanNet) are typically unable to handle out-of-distribution inputs (a known problem in deep learning method).
- Authors also show a surprising result that LSTM are more robust to input noise compared to other methodologies tested, which is at the very least quite an intriguing result.

**Weaknesses:**

- Please do not use $p>0.05$ as evidence to conclude that there is no significant difference (lines 221-230). The only scientific conclusion that can be made when $p>0.05$ is that using this test we *cannot conclude* that a significant difference exists not that there is not a significant difference (emphasis on conclude). Please run a proper statistical equivalence test using which it can be concluded that the two quantities being compared are equivalent.
- Please specify the statistical tests used whose p-values have been reported.
- The argument that there is no difference between the MSE in position between KF, KalmanNet, and LSTM based on p-values > 0.05  is misleading. From figure 2,  it is clear that KalmanNet struggles to estimate the position.
- The error bars for online metrics are missing in Fig 2.

**Questions:**

- Why is the variance of MSE in the position so high in Figure 2?
- Just out of my curiosity, the KalmanNet seems similar in spirit to Extended Kalman Filter, where the Kalman gain is also updated using a known non-linearity in the dynamics of the system. I would like to hear authors' opinion on how extended Kalman filters where the non-linearity can be introduced in the dynamics compare against the methodologies tested in this work, particularly KalmanNet which seems to do something similar where the Kalman gain is updated non-linearly through a GRU. It might provide another avenue for understanding KalmanNet by analyzing if the non-linear trust introduced by the KalmanNet can be modeled as a non-linearity introduced in the dynamics of the system.

**Limitations:**

Yes authors adequately discuss the limitations of the work.

---

> ### Author Rebuttal · Authors · 2024-08-07
>
> We thank the reviewer for highlighting the strengths and main contributions of the paper as well as for suggesting specific avenues for improvement. We have addressed the statistical weakness brought up by the reviewer as well as responding to their questions below.
> ### Weaknesses:
> We thank the reviewer for the various comments and suggestions regarding the statistical analyses of the paper, which we will add in greater detail in a revised main text. The p-values shown in the paper were the result of conducting paired-sample t-tests on the mean squared errors and on the Fisher Z-transform of the correlation values. From the reviewer's suggestion, we also added Bayes paired-sample t-tests for each of the comparisons, to give readers more information about the likelihood of the null hypothesis (models have similar performance) versus the alternative hypothesis (models differ in performance). Specifically, we propose adding the following after the last sentence in line 162, both to specify the statistical test we ran, as well as introduce the new Bayes factor analysis:
>
> >To compare offline performance across models, we used a combination of frequentist and Bayesian statistical methods. First, paired-sample t-tests on the MSEs as well as in the Fisher Z-transformed correlation coefficients [1] determined whether the difference was significant ($\alpha =0.05$) under the null hypothesis assumption. Second, we computed the Bayes factor ($B_{01}$) to determine the ratios of likelihoods between the null and alternative hypotheses.
>
> Additionally, we propose changing lines 221 through 230 to the following:
>
> >In terms of correlation with velocity, which controls the visualization when running online, KalmanNet did not significantly differ from the LSTM ($p = 0.64, B_{01}=3.3$) and had significantly higher correlation than the KF ($p < 1E−7, B_{01}=0.048$) and the tcFNN ($p < 0.001, B_{01}=0.062$; Figure 2, B). All approaches had similar correlations with position, other than the tcFNN, which is a velocity-only approach. Regarding velocity MSE, there was also no significant difference between KalmanNet and the LSTM ($p = 0.72, B_{01}=3.4$) or between KalmanNet and the tcFNN ($p = 0.29, B_{01}=2.1$). However, KalmanNet significantly outperformed the KF ($p < 0.01, B_{01}=0.14$). In terms of position, there were no significant differences in correlation (KNet vs. LSTM, $p = 0.32, B_{01}=2.4$; KNet vs KF, $p=0.93, B_{01}=3.5$; LSTM vs. KF, $p=0.08, B_{01}=1.1$) or MSE (KNet vs. LSTM, $p=0.26, B_{01}=2.0$; KNet vs KF, $p=0.77, B_{01}=3.5$; LSTM vs. KF, $p = 0.1, B_{01}=1.1$), although KalmanNet had higher variance in MSE.
>
> Finally, regarding the error bars representing the standard error of the mean of online metrics in Figure 3: they are present, but are very small in part due to the large number of trials we conducted. The trial numbers can be found in line 234, but for easier access to the reader we propose adding them to the end of the caption, as follows:
>
> >Tested on monkey N across T=601 (KF), 576 (tcFNN), 2801 (KNet), 393 (LSTM) trials in a total of five days.
> ### High MSE variance in Figure 2:
> We thank the reviewer for their question. The higher variance stems from KalmanNet having a bias in the position prediction on some days, which increases the MSE without decreasing the correlation. This effect most likely stems from bias shifts in the neural data, a known problem in brain-machine interfaces [2], which can greatly affect the bias in the linear observation model used in KalmanNet. However, when using KalmanNet online, the virtual hand is driven mostly by predicted velocity, with the predicted position serving only as a stabilization parameter (see equation in line 479).
> ### Similarity to EKF:
> We thank the reviewer for this insightful question. The extended Kalman filter, as well as other varieties such as the unscented Kalman filter, indeed are similar to KalmanNet in the sense that they allow for non-linearities to be introduced either in the dynamics or in the relationship between sensor information and the tracked state. In a linear Kalman filter, the Kalman gain is computed by propagating the noise variance matrices through the linear system. In an extended Kalman filter, on the other hand, since the system is not linear anymore, the Kalman gain is computed by linearizing the system at the predicted state at every time point, and then proceeding as in the linear Kalman filter. This linearization then depends on the state estimate, and the filter can diverge if the system is not linearized around the correct point. In contrast, KalmanNet computes the Kalman gain by implicitly computing the noise of each source of information by using the differences between what it observes and what it predicts (see section A.2.1 of the Technical Appendix). Thus, the difference between the extended Kalman filter and KalmanNet lies in how the noise is propagated through the system, which determines how the Kalman gain is computed. This allows KalmanNet for a much faster switching in which information source to trust, which proved beneficial in this application.
>
> The reviewer also raises an interesting question about whether the non-linear trust introduced by KalmanNet could be modeled as non-linearities introduced in the dynamics. The heteroscedastic Kalman filter (HKF) model presented in the paper presents some evidence towards this: by making the noise on the observations covary with the velocity, the HKF could match the performance with KalmanNet but the HKF’s Kalman gain did not perfectly track the velocity (see Figure 4B). It is not clear whether introducing a non-linearity in the dynamics could make the Kalman gain perfectly track the velocity, but it is definitely an interesting avenue for further understanding KalmanNet and potentially using that to inform model development.
> ### References:
> 1. Meng, X. L., & Rubin, D. B. (1992). Biometrika.
> 2. Degenhart, A. D., …, & Yu, B. M. (2020). Nature BME.

---

> ### Comment · Reviewer_xWsH · 2024-08-08
>
> I have read the review and have elected to keep my original score. I appreciate the extra analysis performed by the authors but they have seemed to missed the point of my comment. I would encourage the authors to read the following statement by the American Society for Statisticians on interpreting p-values, "Wasserstein, Ronald L., and Nicole A. Lazar. "The ASA statement on p-values: context, process, and purpose." The American Statistician 70.2 (2016): 129-133." Particularly, see point 5, "A p-value, or statistical significance, does not measure the size of an effect or the importance of a result." Reporting large p-values does not necessarily indicate that there is no "significant" difference. A large p-value could also result from not having enough samples to run the test with required precision. The english statement of using a large p-value to conclude no significant difference can be misleading and significantly increases chances of mis-interpretation. Let me illustrate through an example. The authors claim that there is no significant difference between the MSE of KalmanNet and LSTM in position but looking at the Fig 2B, it seems more likely that MSE of LSTM is much better than the MSE of KalmanNet. The test used by the authors is most likely not able to produce a smaller p-value simply due to a much larger variance of the MSE of KalmanNet (which is also a bad thing).
>
> The mis-interpretation of p-values is a significant problem in scientific literature leading to hot debates, and as "experts" of machine-learning and statistics, we should do our utmost to not propagate bad-practices regarding interpretation of p-values or statistics in general. I would encourage authors to be very careful when stating conclusions using p-values and carefully translate the "math" to "english"-statements such that they cannot be mis-interpreted.

---

> > ### Author Response · Authors · 2024-08-14
> >
> > We thank the reviewer for their comment and want to note that we completely agree with all the points that they made. We apologize for not being more clear; we originally formulated our results with the underlying assumption of a minimum background of statistics for the average reader, but the reviewer is correct in that explicitly stating the meaning of p-values over the significance level can help the reader interpret our results better. We propose modifying the sentence starting in line 230 to the following:
> >
> > >Overall, it is important to note that the absence of p-values below the significance level for position and velocity correlations and MSE between the LSTM and KalmanNet should not be interpreted as definitive evidence that the two models are equivalent. Instead, it indicates that the data did not provide strong enough evidence to conclude that the models were different. However, the Bayes factor ($B_{01}$) values between $2$ and $3.4$ suggest that the models may have comparable performance, with the null hypothesis (models are equivalent) being at least twice as likely as the alternative hypothesis (models are different). This suggests a higher likelihood that any observed differences are not substantial, though it remains essential to consider the variability and sample size when interpreting these results.

---

### Official Review · Reviewer_JYj7 · 2024-07-12

**Soundness:** 2
**Presentation:** 2
**Contribution:** 3
**Rating:** 5
**Confidence:** 3

**Summary:**

This paper addresses the trade-off between performance and explainability in brain-machine interface (BMI) decoders. The authors introduce KalmanNet, a novel decoding algorithm that combines the traditional Kalman filter (KF) with deep learning techniques, specifically recurrent neural networks (RNNs).

Key contributions:

1. Development of KalmanNet: A hybrid model that maintains the interpretable structure of the KF while leveraging the flexibility of RNNs to compute the Kalman gain dynamically.
2. Comprehensive evaluation: The authors conduct both offline and online experiments using neural data from two non-human primates performing a 2-DoF dexterous finger task. They compare KalmanNet against standard KF, tcFNN, and LSTM decoders.
3. Performance and explainability balance: KalmanNet achieves comparable or better performance than state-of-the-art deep learning models while maintaining a degree of explainability.
4. Behavioral analysis: The paper provides insights into KalmanNet's decision-making process, showing how it adjusts trust between the dynamical model and neural observations.

The paper demonstrates a promising direction for developing BMI decoders that balance high performance with interpretability, potentially enabling safer and more effective neural prosthetics. It also provides valuable insights into integrating control theory and deep learning in the context of neural decoding.

**Strengths:**

1. Originality:
    - The paper presents a novel approach (KalmanNet) that creatively combines traditional control theory (Kalman filter) with modern deep learning techniques (RNNs).
    - Using RNNs to compute Kalman gain dynamically is innovative and addresses a long-standing challenge in BMI decoder design.
    - The introduction of the Heteroscedastic Kalman Filter (HKF) as an analytical tool to understand KalmanNet's behavior is an original contribution.
2. Quality:
    - The experimental design is relatively comprehensive, including both offline and online evaluations, which is crucial for BMI applications. Also, The comparison with state-of-the-art methods (KF, tcFNN, LSTM) provides a robust benchmark for the proposed method.
3. Clarity:
    - The paper is well-structured and follows a logical flow from problem statement to methodology, experiments, and conclusions.
4. Significance:
    - The performance of KalmanNet, matching or exceeding deep learning models while maintaining some interpretability, represents a significant advancement in the field.

**Weaknesses:**

1. Limited generalization: The model's poor performance in new task contexts is a significant weakness. The authors could explore transfer learning techniques to improve the robustness and generalization capabilities of KalmanNet.
2. Insufficient comparison with state-of-the-art methods: The paper lacks comparison with more recent and advanced approaches, particularly transformer-based models which have shown superior performance in various domains. These models also offer various interpretability techniques that could be relevant to this work.
3. Lack of ablation studies: The paper does not provide a detailed analysis of the contribution of each proposed module. Comprehensive ablation studies would help understand the specific gains from different components of KalmanNet.
4. Inadequate analysis of interpretability: While improved interpretability is claimed as a key advantage of KalmanNet, the paper lacks a detailed analysis and concrete examples demonstrating this interpretability in practical scenarios.
5. Questionable paper structure: The absence of a dedicated related work section hinders reviewers' ability to quickly grasp the context of relevant prior work. Additionally, the discussion section is disproportionately long, and some of this space could be better utilized for more detailed experimental analysis and insights.

These weaknesses, if addressed, could significantly strengthen the paper and provide a more comprehensive evaluation of the proposed method in the context of current BMI research.

**Questions:**

1. KalmanNet shows higher sensitivity to injected noise compared to LSTM. Could you provide more insight into why this occurs?
2. Could you provide a more detailed analysis of KalmanNet's sensitivity to hyperparameter choices?

**Limitations:**

The authors have adequately addressed the limitations in the checklist.

---

> ### Author Rebuttal · Authors · 2024-08-07
>
> We thank the reviewer for their very thorough review of the paper, for recognizing its main strengths and contributions, and for suggesting specific avenues for improving our work. We have addressed the weaknesses and questions raised by the reviewer below.
> ### Weaknesses:
> First in terms of a comparison with better state of the art models, we acknowledge that transformer based models have surpassed other models, such as the LSTM, in many applications. However, in previous work [1], with a user in the loop, a transformer that works particularly well for this type of data offline did not perform as well as an LSTM online. This may be most likely explained by the transformer overfitting to the offline brain data and having too complex dynamics for the user to control in a closed loop setting. A recent theoretical study noted that transformers can accurately approximate a Kalman Filter offline [2], but in practice it may not properly discover the domain dynamics from training data alone. Another recent study has shown that transformers may be able to replace the GRUs in KalmanNET [3], which is of strong interest to our group for future work.
>
> Regarding the generalization issue raised by the reviewer, note that while KalmanNet has a big percentage increase in error when testing on a different context, the overall velocity error still falls below that of the Kalman filter and is comparable to the other deep-learning models (see Figure 6). None of these models were optimized for generalization and multiple previous studies have found that generalizing to other tasks in the domain of brain-machine interfaces is a challenging problem [4-5]. In our work, we felt it was important to show that KalmanNet carried over the generalization disadvantages common in non-optimized deep learning models but, as the reviewer suggests, techniques such as transfer learning or data augmentation can have a great impact on the models’ generalization ability and we plan to explore them in future work.
>
> We thank the reviewer for suggesting ablation studies. The original KalmanNet paper (Ref #25 in main) did some ablation work by modifying the architecture and the state-space models, arriving at the architecture we used in our paper. We also did a high-level ablation study of our own by comparing the Kalman filter, KalmanNet, and the heteroscedastic Kalman filter (HKF). The only difference between the KF and KalmanNet is how the Kalman gain is computed, while the only difference between the HKF and the KF is that the noise model changes over time. With our work, we showed that the flexible modulation of the Kalman gain was the key to the better performance in KalmanNet, and we further proved that by emulating that behavior in a linear model. We have also included a new figure (Supplemental figure 2) in the Technical Appendix showing the sensitivity of KalmanNet to the sequence length used during training, also attached to the rebuttal.
>
> We hope that the comments in the general rebuttal on the benefits of explainability help assuage some of the reviewer’s concerns. Additionally, we want to thank the reviewer for suggesting a section specifically about related work: we will condense the discussion and some of the introduction and introduce more details in relevant prior work. Specifically, some of the references we will include, grouped by topics are: Variations on KFs (e.g., adaptive, extended) for BMI applications [6-9]; deep learning models for BMI [5, 10-11] and Refs #16-19 from the main text; and model-based deep learning [12].
> ### Higher sensitivity to injected noise:
> We thank the reviewer for their question. We have addressed this point in the general rebuttal, but briefly: we exposed the models to extreme, out-of-distribution, and non-Gaussian noise, which falls into the worst case for the Kalman filter-based models. The LSTM showing high robustness to this extreme noise was interesting and unexpected and we plan to explore it further in future work.
> ### Analysis of KalmanNet's sensitivity to hyperparameter choices:
> We thank the reviewer for the suggestion. We propose including a more detailed sensitivity analysis on some key parameters during KalmanNet training: length of each sequence, learning rate, and training time. We have included a new figure in the supplement (Supplemental Figure 2) showing the variation in offline MSE across days for different sequence lengths during training, and we will include the learning rate and training time analyses upon publication. Note, however, that the main objective in BMI experiments is to perform well online with a user in the loop, which does not necessarily follow from offline results (Refs #19 and 35 from main text). Online experiments are difficult to perform, necessitating some use of offline data in parameter optimization. We found that the chosen hyperparameters striked a good balance between overfitting to the training data and performing well online.
>
> ### References:
> 1. Costello, J., ... & Chestek, C. (2024). NeurIPS, 36.
> 2. Goel, G., & Bartlett, P. (2024). 6th Annual Learning for Dynamics & Control Conference, 1502-1512.
> 3. Wang, J., Geng, X., & Xu, J. (2024). arXiv preprint arXiv:2404.03915.
> 4. Mender, M. J., ... & Chestek, C. A. (2023). Elife, 12, e82598.
> 5. Temmar, H., ... & Chestek, C. A. (2024). bioRxiv, 2024-03.
> 6. Li, Z., … & Nicolelis, M. A. (2009). PloS one, 4(7), e6243.
> 7. Dangi, S., …, & Carmena, J. M. (2011). 5th International IEEE/EMBS Conference on Neural Engineering, 609-612.
> 8. Tsui, C. S. L., Gan, J. Q., & Roberts, S. J. (2009). Medical & biological engineering & computing, 47, 257-265.
> 9. Malik, W. Q., …, & Hochberg, L. R. (2010). IEEE TNSRE, 19(1), 25-34.
> 10. Pandarinath, C., ... & Sussillo, D. (2018). Nature methods, 15(10), 805-815.
> 11. Sussillo, D., …, & Shenoy, K. (2012). Journal of neural engineering, 9(2), 026027.
> 12. Shlezinger, N., …, & Dimakis, A. G. (2023). Proceedings of the IEEE, 111(5), 465-499.

---

> > ### Comment · Reviewer_JYj7 · 2024-08-14
> >
> > You have resolved some of the concerns I had, and I am inclined to raise my score. However, there are still some issues that need further improvement in the final version.

---

> ### Comment · Area_Chair_7Mju · 2024-08-14
>
> Dear Reviewer JYj7,
> The rebuttal stage deadline is coming soon. Please do not forget to engage in the conversation and let the authors know about your take on their rebuttal, and if appropriate update your score. Thanks for supporting NeurIPS.
> Best,

---

### Official Review · Reviewer_586k · 2024-07-12

**Soundness:** 2
**Presentation:** 2
**Contribution:** 2
**Rating:** 6
**Confidence:** 3

**Summary:**

- This approach studies a few approaches that can be used for neural decoding. The baseline approaches are blackbox DNNs and a vanilla Kalman filter. The proposed approach, the KalmanNet, is a hybrid model, in which a DNN is used to control the gain on a Kalman filter.
- Approaches such as Kalman filtering have the benefit of being more interpretable, at the potential expense of performance. Despite this they find comparable results with pure deep learning approaches.
- Other aspects of the trade-off between the "traditional" KF approach and the DNN approaches are explored.
- By examining the KalmanNet's predicted gain, the authors can measure at which time points the KalmanNet relied more on observation and when it relied more on the prior.

**Strengths:**

- The paper is well written; it is of general interest to the BMI community to see the results of exploring this tradeoff
- The benchmarked models are relevant to the ones currently popular in the BMI field
- There are specific takeaways for future BMI decoding models given in the conclusion
- Data for evaluation seems to have been newly collected for this study (is this indeed the case?) which represents significant novelty

**Weaknesses:**

- Overall, I think the technical novelty is a little limited. That is, the application of KalmanNet is a good engineering contribution, but the presented modifications to the KalmanNet seem mainly to adapt an existing model to this particular decoding task.
- The main contribution of this paper seems to be an empirical comparison of existing methods on a specific neural decoding problem. This might have limited significance to the broader machine learning community.
- It's mentioned that the KF has a safer operation (line 268). Can this still be said of KalmanNet, since the contribution of KF can be zeroed out at anytime by the network?
- If I read section 3.3 correctly, it seems that the KalmanNet is not as robust to injected noise as the LSTM, which is unfortunate. I don't think the authors should be penalized for disclosing this result. They should be commended. But it does hurt the significance of the proposed modified KalmanNet approach, which we might have expected to be more robust to noise.

**Questions:**

- Line 93 mentions dynamics model $A$, but this does not appear in equation 1. Is it supposed to?
- What do online and offline refer to? Simply the presence of ground truth finger measurements?
- A basic question: what is the difference between modifying the Kalman gain using a network and building a black box network that takes the observations directly? Is it a difference in expressive power? Is it a difference in safety guarantees?
- A related question: what are the benefits of explainability in this context? What are some things that we get for knowing that we are relying on the observations?
- line 223: What is $p$? Pearson's $r$ correlation? The significance of the difference between approaches?
- Did any modifications to KalmanNet need to be made to accomodate this domain?
- Figure 5: Show the whole product of duration and magnitude?

**Limitations:**

Limitations are discussed adequately. Negative societal impacts are not discussed, but I think this is less relevant here.

---

> ### Author Rebuttal · Authors · 2024-08-07
>
> We thank the reviewer for their detailed comments and suggestions, as well as for recognizing the main strengths of the paper. We have addressed the paper weaknesses brought up by the reviewer in the general rebuttal. Here, we will address each of the reviewer’s questions.
> ### Equation 1:
> We thank the reviewer for noticing this oversight. We propose modifying the referred sentence to the following:
>
> >The trainable parameters for the KF correspond to the linear observation model (C), the linear dynamics model, and the noise covariances of the state and the observations.
> ### Online and offline experiments:
> We thank the reviewer for this question, which allows us to expand on one of the paper's key contributions. We refer to offline experiments as those in which we analyzed offline the brain activity and finger kinematics from the monkey doing the task in hand control. For these, we have the ground truth readings of the finger measurement, as the reviewer stated, which means we can determine the error in predictions by each model. Offline analysis of neural data is a domain in which these neural network tools are widely used and have very high performance.
>
> In online experiments, there is a user in a control loop reacting moment by moment to how well the output movements match their desired movements. This is the domain for which Kalman filters are much more commonly used than neural networks due to their physically stable, easily controllable dynamics. In our application, we have the monkey control animated fingers with his brain signals in real-time using one of the tested decoders. We feed those signals every 50ms to the model we are testing and use that model to predict the desired finger velocity and then we move the virtual hand based on those predictions. The monkey sees the movement of the virtual hand and reacts to try to acquire targets as fast as possible. Offline performance is not necessarily predictive of online performance (Refs #19, 35 from main). The user’s reactions to movement at each timestep can generate brain activity that looks very different from the training data, consequently affecting the predictions. We saw this effect in this paper with the tcFNN model. Thus, our work included the very essential online experiments, beyond just offline modeling, to validate that the tested models would work in a real-life scenario.
>
> To clarify the difference between these two experiment modalities, and relate them to the descriptions of hand and brain control, we propose adding a sentence at the end of the paragraph starting on line 129:
>
> >Analyses of performance during hand and brain control trials are referred to as offline and online, respectively.
> ### Kalman gain with a network vs a black box:
> We thank the reviewer for this question. The biggest difference between modifying the Kalman gain using a network and building a black box network that directly takes the observations lies in the framework in which KalmanNet operates. In KalmanNet, the network does not predict the velocities directly but rather just determines at every time point how much to trust the linear observation model versus how much to trust the linear dynamical model. This allows KalmanNet to incorporate domain knowledge in the state-space model, a classic advantage of using the Kalman filter in any application. In our case, this domain knowledge is reflected by our dynamics model, which models the physics of finger positions and velocities, and the observation model, which determines the relationship between brain measurements and finger kinematics, both of which have been informed by previous work (Ref #31, 36 in main). Thus, and also for safety, KalmanNet essentially never directly zeroes out the contribution of the Kalman filter, but rather adds additional flexibility to the model to choose whether to trust dynamics or sensor measurements at any time point.
> ### Benefits of explainability:
> Please see the general rebuttal for a thorough response to this question.
> ### p, Pearson's r correlation:
> In line 223, the p represents the p-value of the difference between correlations. The value of 0.64 means we did not find enough evidence to conclude that the two models had different correlations. On the suggestion of another reviewer, we have also added a computation of the Bayes factors, to determine the ratio of likelihoods for the null versus alternative hypothesis in each comparison (please see response to reviewer xWsH for full explanation).
> ### Domain modifications to KalmanNet:
> We thank the reviewer for this question. Our novel modifications to make KalmanNet work in this domain can be separated into two parts: the state-space model and the training. The biggest difference between the original KalmanNet and the one used in this work is that we created a new state-space model (dynamics and observation models) based on the specific domain of application (finger movements and brain sensor measurements). The state-space model used the structure described in section A.1 of the Technical Appendix, but briefly: we used a kinematic model of the physical position dynamics via velocity integration, and learned from the training data the parameters for the velocity dynamics as well as the relationship between observations and kinematics. For model training, we modified the base loss function to account for the different scales for our two predictors, position and velocity, and increased the input sequence length to improve performance and encourage smoothness in the output. To show this, we will include a new figure that shows the change in velocity MSE with different sequence lengths during training to the Technical Appendix, as seen in the attached PDF (Supplemental Figure 2).
> ### Figure 5:
> We thank the reviewer for the suggestion. We propose adding the values from the product of duration and magnitude as a figure to the Technical Appendix (Supplemental Figure 3), shown in the attached PDF.

---

> > ### Comment · Reviewer_586k · 2024-08-13
> > **Response to author rebuttal**
> >
> > I thank the authors for taking the time to answer my questions. I will keep my score and continue to recommend acceptance. I now better understand the explainability argument: the KalmanNet filter only controls a tuning parameter in the KF. Overall, I think the technical work represented by this contribution is thorough, but possibly of limited significance. The proposed state space model and training scheme seem fairly specific to this application. But I don't think these are strong reasons to reject, since it may be of some interest to the BCI community.

---

### Author Rebuttal · Authors · 2024-08-07

We thank the reviewers for their helpful questions, suggestions, and generally supportive comments. Our work demonstrates the tradeoffs of using KalmanNet, an explainable algorithm that combines deep learning with the Kalman filter (KF), to predict finger movements from brain data. We have addressed the weaknesses and questions raised by the reviewers here and in each review. We believe that with the addition of the proposed modifications this work is greatly improved.
### Novelty and significance:
We appreciate the reviewers recognizing our novel approach and comprehensive evaluation of our decoders in online BMI tasks with newly collected data. We wish to clarify that, yes, all data collected during online tasks are new data specifically for this project and used for comparing the real-time performance of the 4 decoders, 2 of which are new (KalmanNet, HKF). Additionally, 3 of the 13 days used for offline evaluation were new data collected specifically for this project. Upon publication, we will release all data for public usage.

We will revise the explanation of our novel approach in the text to clarify that the primary novelty is in combining a state space model framework and a deep learning architecture for BMI applications. However, we also suggest that these results are significant for any controls application with a user in the loop, interacting with the physical world. KFs are widely used due to safety concerns in these applications, which may limit performance compared to state of the art machine learning techniques. Our investigation of the behavior of KalmanNet during the finger movement task led us to create a new, explainable, and small linear model: a novel and significant advance for models with BMI applications. Thus, we show the benefit of potentially understanding the algorithms “mechanisms”, which is not usually available for black-box models. Additionally, we demonstrated that explainable models do not necessarily perform worse than their pure ‘black-box’ counterparts with novel online experiments.  Finally, we show an intriguing result in which an LSTM substantially outperformed both a normal KF and KalmanNet in the case of extreme noise injection, which was counter to our original intuition.
### Explainability:
Some reviewers raised concerns about the benefits of KalmanNet’s explainability. First, throughout robotics and controls, engineers generally choose “white-box” (explainable) approaches at the cost of performance, due to safety concerns stemming from “black box” approaches. This is important not only for brain machine interfaces which control a prosthesis or muscle stimulation, but for any robotics application interacting with the physical world. Autonomous vehicles make greater use of KFs than deep learning models, despite possible higher performance. Second, explainability is helpful for refining the design of algorithms. After observing KalmanNet’s behavior, we generated a simpler linear decoder (HKF), with only a fraction of parameters, that matched the performance of the deep learning model. Third, this result may even shed light on the mechanisms by which the brain drives motor output. KalmanNet learned to trust the brain for producing high velocities and trust dynamics for stopping. This is consistent with prior work in systems neuroscience recording from the motor cortex [1,2].
### Noise injection:
Reviewer 586k pointed out that we should not be penalized for disclosing the fact that KalmanNet has a  weakness to extreme noise. Reviewer xWsH shared our view that this was a very intriguing result. Indeed, we performed this analysis because intuitively, one would expect the KF to do well in the face of noisy inputs, where it can rely on its dynamical model and avoid unsafe movements. The fact that the LSTM outperformed both KalmanNet and a regular KF in this regime is a very interesting result in our opinion.

This occurs because the KF architecture is subject to an underlying assumption of zero mean Gaussian noise and is known to suffer when this assumption is violated [3-4]. With KalmanNet, although we do not explicitly model the noise, the model is subject to the same assumptions. We exposed these models to extreme levels of noise, up to 100x the standard deviation of the neural data. We want to show this data for two reasons: First, neural signals are very small, and 100x errors are possible. Second, we want to highlight the surprising success of the LSTM for this difficult problem.

We have now included another figure (Supplemental Figure 1) that compares the MSE for velocity predictions when the noise is less extreme (up to 1x standard deviation). There, the difference between models is small, suggesting that when operating within the expected noise magnitudes, KalmanNet can match the LSTM performance. We will rephrase (line 273):

>Given the small magnitude of brain signals, noise artifacts can be much larger than the signal features of interest. We modeled those with extreme additions of noise ~100 times the variance in the training data.

Pragmatically, this low robustness to out-of-distribution noise is also a known weakness of the KF and many techniques have been developed to address it. For example, the outlier insensitive KF (OIKF) [5] explicitly models outliers as random variables with unknown variance. The outlier robust KF (ORKF) [4], modifies the noise model to allow for non-Gaussian and heavy-tailed noise. Since KalmanNet works under the same framework, we could apply these techniques to improve KalmanNet’s robustness to noise.

### References:
1. Saleh, M., …, Hatsopoulos, N. G. (2010). Journal of Neuroscience, 30(50), 17079-17090.
2. Reina, G. A., ..., Schwartz, A. B. (2001). Journal of neurophysiology, 85(6), 2576-2589.
3. Huber, P. J. (1992). Breakthroughs in statistics: Methodology and distribution, 492-518.
4. Agamennoni, ..., Nebot, E. M. (2011). IEEE ICRA, 1551-1558.
5. Truzman, S., ..., Klein, I. (2023). ICASSP, 1-5.

---

### Decision · Program_Chairs · 2024-09-25

**Decision:**

Accept (poster)

**Comment:**

Range of rating was only 1 point difference suggesting very good agreement between reviewers. Rounded average score was 6 (accept)
Something in which all reviewers agree and I do as well is that this paper is very well written. Moreover, all reviewers agree on an accepting score. I could see myself having been a bit more tight here as I believe the paper is exploiting a bit of opportunism, but I do not want this comment of mine to detract from the fact that it is indeed a good paper. There was some criticisms wrt the interpretation of stats, and again I agree with the reviewers on this regard, but as proven by the favourable scores this was not perceived as critical.
Rebuttal was nicely done by the authors but discussion was hopeless. I intervened asking participation from one of the reviewers who seemed disengaged, but even that did not seem to have triggered a more spicy argumentation. Anyhow, I reckon everyone was aware that this paper was a clear acceptance and hence not worth spending more time on defending (or yielding) positions.
No issues with ethics were highlighted.